# Controlling the bioactivity of a peptide hormone in vivo by reversible self-assembly

Myriam M. Ouberai[1], Ana L. Gomes Dos Santos[2], Sonja Kinna[1], Shimona Madalli[3], David C. Hornigold[3], David Baker[3], Jacqueline Naylor[3], Laura Sheldrake[4], Dominic J. Corkill[5], John Hood[6], Paolo Vicini[6], Shahid Uddin[2], Steven Bishop[7], Paul G. Varley[2] & Mark E. Welland[1]

The use of peptides as therapeutic agents is undergoing a renaissance with the expectation of new drugs with enhanced levels of efficacy and safety. Their clinical potential will be only fully realised once their physicochemical and pharmacokinetic properties have been precisely controlled. Here we demonstrate a reversible peptide self-assembly strategy to control and prolong the bioactivity of a native peptide hormone in vivo. We show that oxyntomodulin, a peptide with potential to treat obesity and diabetes, self-assembles into a stable nanofibril formulation which subsequently dissociates to release active peptide and produces a pharmacological effect in vivo. The subcutaneous administration of the nanofibrils in rats results in greatly prolonged exposure, with a constant oxyntomodulin bioactivity detectable in serum for at least 5 days as compared to free oxyntomodulin which is undetectable after only 4 h. Such an approach is simple, cost-efficient and generic in addressing the limitations of peptide therapeutics.

[1] Nanoscience Centre, Department of Engineering, University of Cambridge, Cambridge CB3 0FF, UK. [2] Biopharmaceutical Development, MedImmune Ltd., Granta Park, Cambridge CB21 6GH, UK. [3] Cardiovascular and Metabolic Diseases, MedImmune Ltd., Granta Park, Cambridge CB21 6GH, UK. [4] Invivo Sciences UK, MedImmune Ltd., Granta Park, Cambridge CB21 6GH, UK. [5] Respiratory, Inflammation and Autoimmunity, MedImmune Ltd., Granta Park, Cambridge CB21 6GH, UK. [6] Clinical Pharmacology, Drug Metabolism and Pharmacokinetics, MedImmune Ltd., Granta Park, Cambridge CB21 6GH, UK. [7] Biopharmaceutical development, MedImmune LLC Gaithersburg Headquarters, One MedImmune Way, Gaithersburg, MD 20878, USA. Correspondence and requests for materials should be addressed to M.E.W. (email: mew10@cam.ac.uk)

Peptides have unique potential as a basis for the development of therapeutic agents owing to their high potency and specificity and their excellent safety profile. In 2013, clinical trials of 128 peptides demonstrated their ability to treat a broad range of conditions, including cancer, autoimmune, cardiovascular, and metabolic diseases[1, 2]. Although there are a number of peptides on the market their full potential as therapeutic agents remains a significant challenge due to their low solubility, chemical instability, aggregation propensity, low stability against proteases, high clearance, and short duration of in vivo activity[2, 3]. Strategies to chemically modify peptides are therefore being developed from native peptide sequences to optimize peptide stability and residence time in serum. One of the common ways of improving peptide bioavailability is to engineer peptide analogs containing unnatural amino acids and conjugate them to fatty acids, polymers, or large proteins[4–6]. Considering that subcutaneous (s.c.) injection is the most common administration route for peptides, reducing the injection frequency and achieving consistent and accurate delivery are requirements for treatment adherence and patient comfort—especially in the effective management of chronic diseases. To this end, the pharmacokinetic properties of peptide analogs can be optimized for long-term controlled release using polymer-based formulations (e.g., the marketed formulations Bydureon®, Zoladex®, and Lupron Depot®)[7].

A very different strategy of addressing the limitations above is to use naturally occurring processes such as supramolecular self-assembly, by which many polypeptides form highly ordered and stable nanostructures[8–13]. This remarkable generic process, which is exploited by living systems, is based on the intrinsic propensity of polypeptide chains to self-assemble into β-sheet rich amyloid-like fibrils via a hydrogen bond network[14]. For instance, various peptide hormones including glucagon are stored in the form of amyloid-like nanofibrils in secretory cells[15], and the concept of an amyloid depot has been applied to the design of an analog of the gonadotropin-releasing hormone degarelix available in the market as Firmagon®[16]. Upon s.c. administration in both rodents and humans, degarelix forms a depot attributed to the formation of amyloid-like fibrils from which the peptide is released into the systemic circulation[16, 17]. Yet, in spite of this very particular example, based on a short (10 amino acids) and chemically modified peptide sequence lacking secondary structure, the broad application of self-assembling nanofibrils as a strategy to address the limitations of peptide therapeutics still needs evidence of the propensity of native peptides to be formulated as reversibly self-assembling nanostructures; the release of active peptide from the nanofibrils which recovers its initial conformation and produces a pharmacological effect; the sufficient in vivo release of active peptide from the nanofibrils to achieve the desired pharmacokinetic profile.

In this study, we investigate a native peptide, oxyntomodulin (Oxm), which is a 37-amino-acid proglucagon-derived peptide hormone with sequence homology to both glucagon and glucagon-like peptide-1 (GLP-1)[18]. As a dual agonist of both glucagon and GLP-1 receptors, Oxm has been shown to be a promising pharmacological agent in the treatment of obesity as it suppresses food intake and increases energy expenditure in both rodents and humans[19–22]. Native Oxm has the advantageous property of simultaneously achieving glucose control and weight loss and consequently is an attractive therapeutic agent to treat both diabetes and obesity[23]. However, the clinical impact of Oxm is currently limited due to its short elimination half-life of 12 min in humans, a limitation which is currently being addressed by complex engineering of analogs. For instance, substantial modification of the Oxm primary sequence is necessary to reduce proteolytic degradation and increase in vivo circulation time[24–27]. This complicated approach to engineer the native peptide sequence in order to both retain activation of glucagon and GLP-1 receptors together with a prolonged residence time in serum is challenging. For example, the recent finding that Oxm analogs increased food intake in rats, exactly the opposite of what is being targeted, illustrates the complexity of achieving the desired therapeutic effect based on peptide sequence modification[28].

We take a much simpler approach. As Oxm contains the 29-amino acid sequence of glucagon we expected it to self-assemble into nanofibrils[19, 29]. Here we show that Oxm readily self-assembles under mild conditions into nanofibrils having intrinsic long-term stability, which can be controlled to subsequently dissociate to release intact and active peptide under physiological conditions. We characterize the reversible self-assembly process using a unique suite of complementary techniques and assays, and describe the in vitro and in vivo bioactivities of fibrillar and released peptides, as well as the pharmacology and pharmacokinetic profile of fibrillar Oxm. After s.c. administration of the nanofibrils in rats, Oxm bioactivity in serum is significantly prolonged with the equivalent of 1 nM Oxm present in serum for at least 5 days after a single s.c. injection of 10 mg kg$^{-1}$. Following administration of the same dose of free Oxm, serum Oxm bioactivity is undetectable after 4 h. We also show that s.c. administration of fibrillar Oxm in mice produces a pharmacological effect on glucose lowering. These studies demonstrate the potential of reversible self-assembly to reduce dose frequency while maintaining efficacy for peptide therapeutics used to treat chronic diseases such as obesity and diabetes.

## Results

**Oxm self-assembles into fibrillar nanostructures.** Self-assembly of Oxm was carried out at a concentration of 10 mg mL$^{-1}$ in water at an incubation pH of between 7.0 and 7.3 and at a low

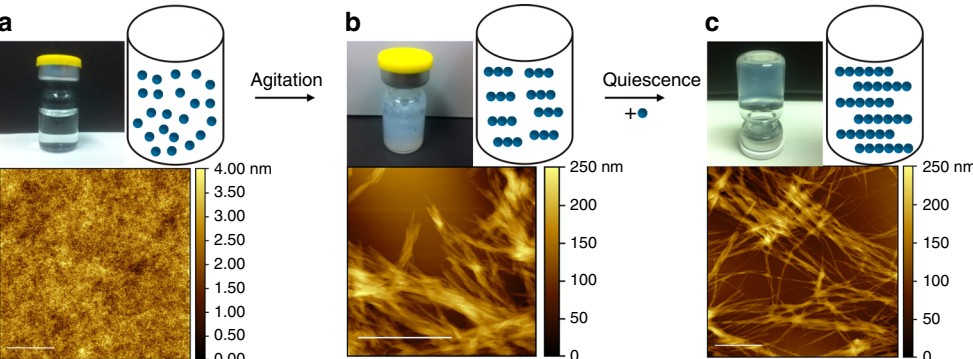

**Fig. 1** Self-assembly of Oxm. **a–c** Photographs, schematic representations and AFM images of freshly prepared peptide at 10 mg mL$^{-1}$ in 0.09% saline **a**, after 5 days of incubation with orbital shaking **b** and after incubation of 0.1 mg mL$^{-1}$ fibrils with a 10 mg mL$^{-1}$ free peptide solution in water **c**. Scale bar, 1 μm

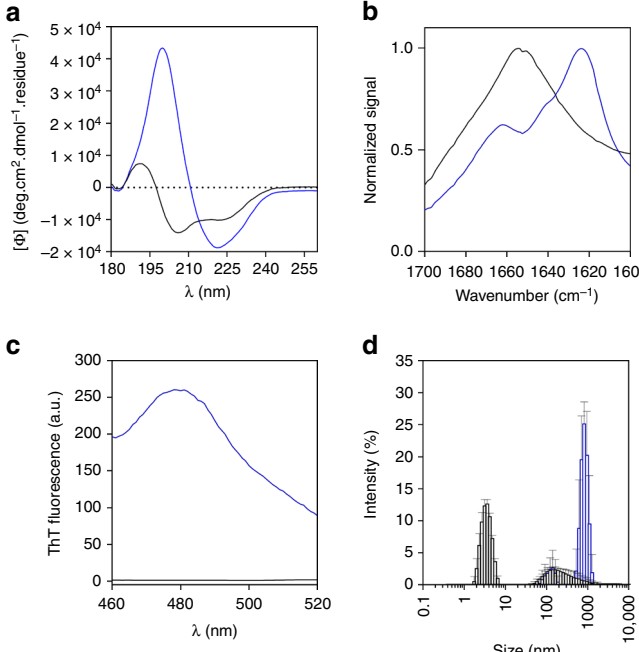

**Fig. 2** Structural properties of free and fibrillar Oxm. **a–d** Free (black) and fibrillar (blue) Oxm at 1 mg mL$^{-1}$ in 0.09% saline. Far-UV CD **a**, ATR-FT-IR, **b** and ThT emission spectra **c**, and DLS analysis (Error bars represent standard deviations obtained from eight measurements of the same sample) **d**

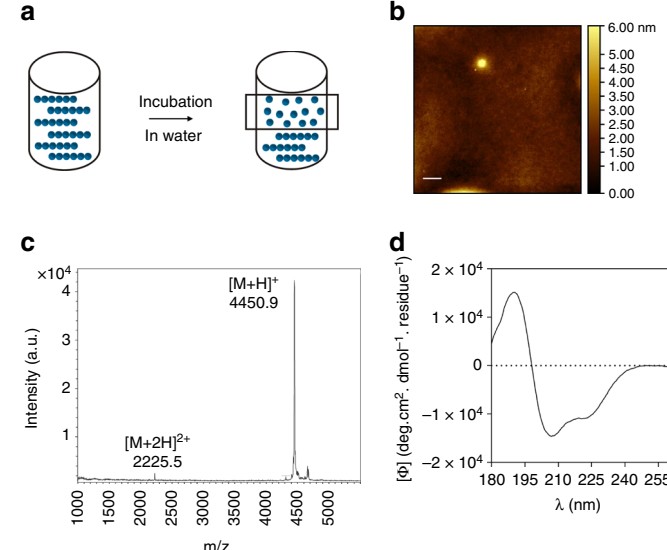

**Fig. 3** Characterization of released peptide. **a–d** Schematic of the stability study **a** with examples of a representative AFM image **b**, mass spectrum **c** and far-UV CD spectrum **d** of released Oxm in water. Scale bar, 1 μm

ionic strength (0.09% saline), in order to screen the positive charge (approx. 3+) of the peptide. In order to further promote self-assembly, the solution was incubated at 37 °C and agitated by orbital shaking. The freshly prepared solution was clear, and atomic force microscopy (AFM) analysis showed a homogeneous and smooth layer of peptide that was deposited onto mica (Fig. 1a). No noticeable aggregates were visualized. After incubation for 5 days, the solution turned turbid due to the formation of a suspension of aggregates (Fig. 1b). AFM analysis showed the presence of fibrils with ~10 nm diameter and up to 1 μm length, grouped into bundles. The conversion yield of the self-assembly of Oxm into fibrillar nanostructures was estimated to be 99% under these conditions.

The nanofibrils were next used to seed a solution of free peptide at 10 mg mL$^{-1}$ in water. The solution was incubated without agitation for 1 week and then diluted to 1 mg mL$^{-1}$ in 0.09% saline for another 2–9 days of incubation at 37 °C and then 9 days at room temperature. Most of the free peptide in the solution was converted into nanofibrils with a conversion rate of 94%. The solution took on a gel-like consistency, and the AFM analysis showed a network of nanofibrils with diameters of ~10 nm and lengths between 2 and 9 μm (Fig. 1c).

Structural analysis of free and fibrillar Oxm was performed using far-UV circular dichroism (CD) (Fig. 2a and Supplementary Fig. 1). Free peptides were mainly arranged in an α-helical conformation (36% α-helical and 12% β-sheet content). In contrast, the far-UV spectrum of fibrillar Oxm showed a significant change in the secondary structure, with 57% β-sheet and 14% α-helical content.

A conformational change of the peptide was also assessed by attenuated total reflection (ATR) Fourier transform infrared (FT-IR) spectroscopy (Fig. 2b). The analysis of the ATR-FT-IR data suggests that free Oxm is composed of disordered and α-helix structures as indicated by the band at 1650–1655 cm$^{-1}$[30, 31]. By contrast, fibrillar Oxm displays a maximum at 1624 cm$^{-1}$ as well

as a prominent shoulder at 1641 cm$^{-1}$, which indicate that peptide conformation is composed of β-sheet structure. The presence of a band at 1662 cm$^{-1}$ can indicate the presence of β-turn or $3_{10}$-helix structures[30, 31].

When the probe thioflavin T (ThT) was applied to fibrillar Oxm, a characteristic fluorescence emission at 480 nm confirmed that the peptide had adopted a cross-β-sheet conformation (Fig. 2c)[32]. As expected, no emission was observed in the presence of freshly prepared Oxm. Finally, dynamic light scattering analysis of free and fibrillar Oxm showed a significant shift in the size of species present in solution (Fig. 2d). For the free peptide solution, the size distribution shows a major sub-population of species with a mean diameter of 3.6 nm. In contrast, the fibrillar Oxm sample contained significantly larger species, with a mean diameter of 825 nm. Altogether, we show that Oxm can be converted, under mild conditions and with a high yield, into fibrillar nanostructures displaying amyloid-like features.

**Oxm nanofibrils dissociate to release intact peptide**. To assess the stability of fibrillar Oxm, 1 mg mL$^{-1}$ nanofibrils were incubated in five media: phosphate buffer (25 mM, pH 7.5), Tris-HCl buffer (25 mM, pH 7.5), 0.09% saline, water and aqueous HCl (10 mM, pH 2). After 48 h in 0.09% saline, fibrillar Oxm was stable and no release of peptide was detected from the fibrils (Supplementary Table 1). As the stability of hormone fibrils has been described to depend on the presence of salts of monoprotic or polyprotic acids[33], the stability of fibrillar Oxm was assessed in phosphate and Tris-HCl buffers. In both buffers, fibrillar Oxm was stable with less than 1.5% of released peptide detected after 48 h. By contrast, in water 37% of peptide was released after 4 h incubation, and 53% was released by 48 h (Supplementary Table 1). Moreover, in aqueous HCl, a 77% release was observed after only 4 h. This shows that electrostatic interactions play a major role in the stability of fibrillar Oxm—the positive charge of the peptide at both acidic and neutral conditions induced an electrostatic repulsion between peptide molecules, which rendered the fibrillar state unstable. The presence of salts such as NaCl, phosphate or Tris-HCl stabilizes the fibrillar state at 1 mg mL$^{-1}$. We therefore anticipate that different equilibrium states are reached depending on the solution conditions, such as the peptide concentration, the presence of various salts and pH.

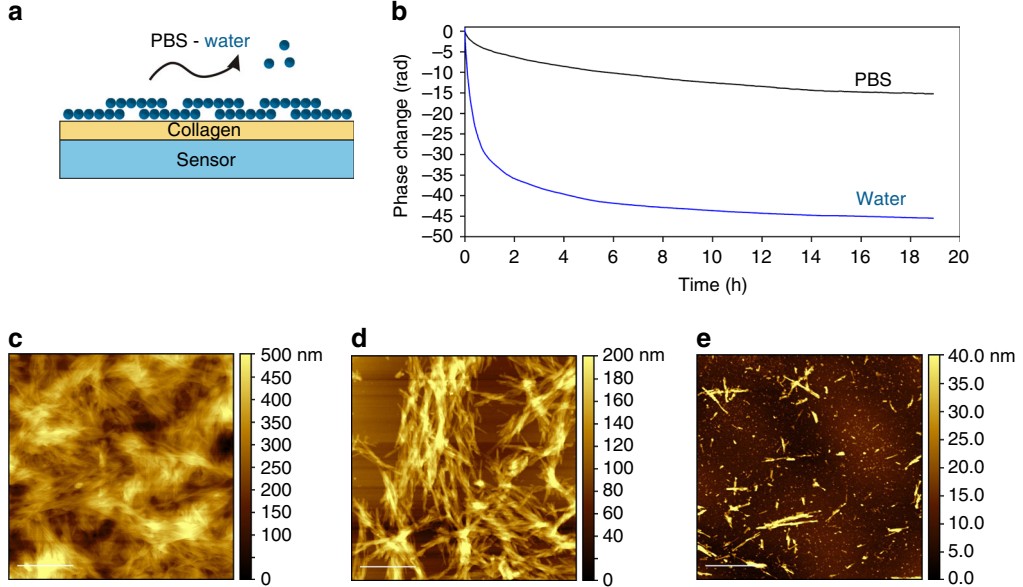

**Fig. 4** Dissociation profile of Oxm nanofibrils under physiological conditions. **a** Schematic of the dissociation study using the DPI biosensing technique. **b** Phase changes as a function of time on top of the DPI sensor coated with collagen and fibrillar Oxm in water (blue) and PBS (black). **c–e** Representative AFM images of Oxm nanofibrils deposited onto a collagen layer before **c** and after incubation in PBS **d** and water **e**. Scale bar, 1 μm

Investigations to further characterize the equilibrium properties of the system under various conditions are ongoing.

The AFM images of released peptide in water show a smooth layer of peptide spread onto mica (Fig. 3a, b). As measured by mass spectrometry, the peptide remained chemically intact after release from the nanofibrils ($[M + H]^+ = 4450.9$, Fig. 3a, c). Furthermore, the far-UV CD spectrum of released Oxm in water shows that the peptide recovered its initial conformation, which is characterized by helical and disordered structures in proportions similar to free peptides (Fig. 3a, d and Supplementary Fig. 2). These results highlight the ability of self-assembled peptide nanofibrils to release intact peptides that can fold from a cross-β-sheet structure back to their initial helical conformation, which is a key structural element for the activation of cell receptors[34].

A surface-based technique, dual polarization interferometry (DPI), was used to assess the dissociation profile of Oxm nanofibrils under conditions mimicking physiological pH, ionic strength and temperature, with the fibrils in interaction with one main component of the s.c. space, and with peptide cleared upon release from the fibrils. DPI is one of the most powerful label-free biosensing techniques available; it can be used to monitor the real-time phase changes resulting from the variation in thickness and refractive index of layers deposited on the top of a sensor[35, 36]. In this study, DPI was used to characterize the dissociation process of fibrillar nanostructures in conditions mimicking the s.c. compartment (with continuous flow to clear released peptide upon fibril dissociation and fibrils deposited onto a collagen layer; Fig. 4a)[37]. A decrease in phase changes as a function of time is observed under these conditions, indicating the removal of material from the surface, with a sharper decrease during the first 4 h in water than in phosphate-buffered saline pH 7.4 (PBS; Fig. 4b). AFM images of the surface before (Fig. 4c) treatment, and after flowing with PBS (Fig. 4d) or water (Fig. 4e) for 20 h, show that the removal of materials corresponded to a dissociation of the nanofibrils. This result is indicated visually by less fibril coverage on the surface of the sensor, with only a few residual short fibrils imaged after incubation in water. Quantification of the change in thickness, density and mass after 4 h incubation shows that 42% of the deposited material was removed from the surface in water, whereas only 10% was

removed in PBS (Supplementary Table 2 and Supplementary Fig. 3). Therefore, under physiological conditions in PBS, Oxm nanofibrils dissociate with a sustained release profile. This finding contrasts with the trend observed for amyloid-like fibrils, which are known to be stable under various solution conditions. For instance, glucagon fibrils are generally stable even under the harsh conditions of thermal and chemical denaturation, but their stability appears to be related to the structure of fibrils and self-assembly conditions (salts, peptide concentration and temperature)[38]. In those experiments, the trend observed implied that complete dissociation of glucagon fibrils can occur upon high dilution, given time.

**The released peptide is active and nontoxic in vitro.** Having shown the chemical stability of the released Oxm, we assessed its functional potency in comparison to that of free and fibrillar Oxm. Agonist potency determination for all samples was performed using cAMP accumulation assays in Chinese hamster ovary (CHO) cells expressing recombinant human GLP-1 (hGLP-1R) or glucagon (hGCGR) receptors (Fig. 5a, b). Free, released and fibrillar Oxm all acted as full agonists in hGLP-1R- and hGCGR-expressing cells, compared to the maximum effects of GLP-1 or glucagon peptides in GLP-1R and GCGR assays, respectively.

Released and free Oxm had comparable potencies against hGLP-1R, with $EC_{50}$ values of $96.6 \pm 31.0$ (geometric mean ± s.e.m.) and $72.4 \pm 17.9$ pM, respectively, whereas fibrillar Oxm was approximately 40-fold less potent with an $EC_{50}$ of $2938.3 \pm 655.7$ pM (Supplementary Table 3). In the hGCGR-expressing line released and free Oxm also showed similar potencies, with $EC_{50}$ values of $26.0 \pm 5.2$ and $18.2 \pm 4.2$ pM, respectively, whereas fibrillar Oxm was ~62-fold less potent, $EC_{50}$ $1120.5 \pm 302.2$ pM (Supplementary Table 3). Formation of nanofibrillar structures is expected to preclude pharmacological activity of the peptide, therefore any measured in vitro activity of the fibrillar material is likely to be due to release of small amounts of non-fibrillated peptide in the assay conditions.

To assess the in vitro cytotoxicity of Oxm species, a cell viability assay was performed in the presence of free, released, and fibrillar Oxm. Metabolic bioactivity of living cells was measured

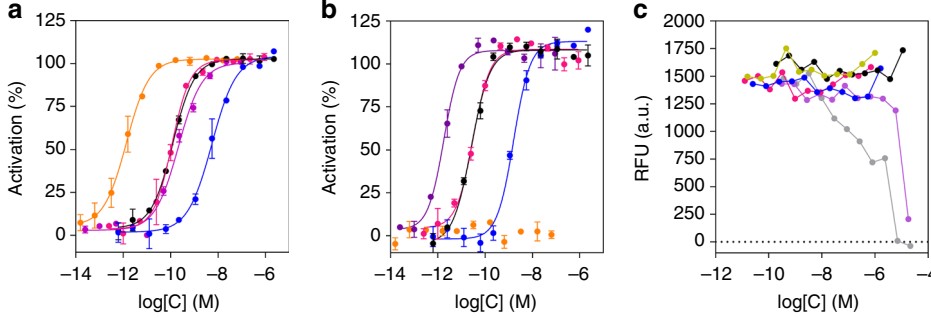

**Fig. 5** Agonist potency and cytotoxicity profiles of Oxm species. **a**, **b** In vitro potencies determined in cAMP accumulation assays in CHO cell lines expressing human GLP-1 **a** and GCG **b** receptors of free Oxm (black), released Oxm (magenta), fibrillar Oxm (blue), glucagon (violet), GLP-1 (orange). Data show representative curves of >5 independent experiments. Curve data are the arithmetic mean ± s.d. of duplicate data points. **c** 48 h cytotoxicity prolife in CHO-GLP-1R cells of vehicle (yellow), free Oxm (black), released Oxm (magenta), fibrillar Oxm (blue), Ro-318220 (violet), Staurosporine (gray). RFU = Relative fluorescence units. Data show representative curves of three independent experiments

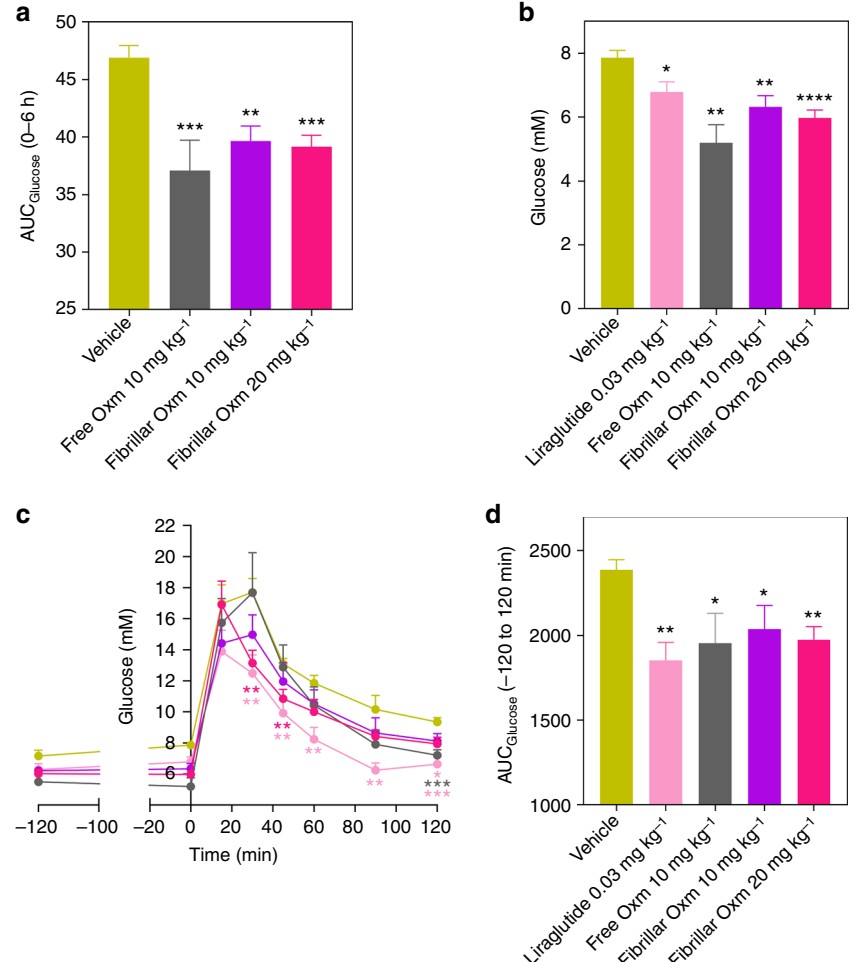

**Fig. 6** Effect of fibrillar Oxm on blood glucose lowering. Glucose lowering demonstrated after 6 h of administration with either vehicle or free and fibrillar Oxm by either AUC **a**, or blood glucose concentration at 6 h **b**. A glucose tolerance test (GTT) was conducted at 6 h post-dose **c** and efficacy was also expressed by AUC **d**. Liraglutide was used as positive control and administrated 2 h prior to GTT. Data are mean ± s.e.m. ($n$ = 7–8 animals per group). $*P < 0.05$, $**P < 0.01$, $***P < 0.001$, $****P < 0.0001$, vs. vehicle, two-sided, unpaired $t$-test

through bioreduction of a resazurin-based dye (Fig. 5c). None of the forms of Oxm were cytotoxic at concentrations up to 1000-fold above the $EC_{50}$ value in the CHO-hGLP-1R cell line. Two positive control cytotoxic agents, Ro-318220 and staurosporine, both protein kinase C inhibitors, showed the expected curves of concentration vs. loss of cell viability. These studies show that the

fibrillar state of Oxm is not acutely cytotoxic at the concentrations tested and can be used as a reservoir in which active and nontoxic peptides can be stored and released upon dissociation of the nanofibrils under physiological conditions. Our result is in agreement with studies describing mature fibrils as harmless reservoirs of polypeptides with the toxic species being the non-

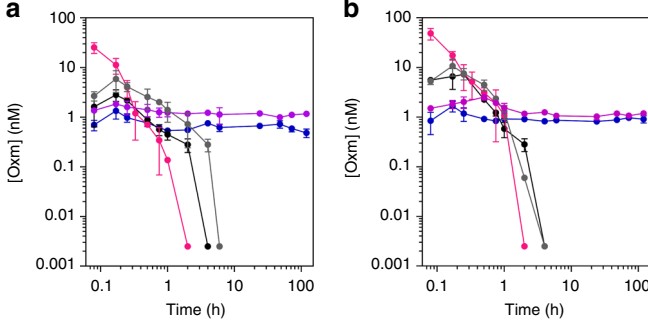

**Fig. 7** Pharmacokinetic profiles following administration of free and fibrillar Oxm. Oxm bioactivity in rat serum determined using in vitro cell-based cAMP bioassay for determining GLP-1 **a** and GCG **b** receptor agonist bioactivity after administration of 5 mg kg$^{-1}$ of free Oxm i.v. (magenta), s.c. (black), and fibrillar Oxm s.c. (blue) and 10 mg kg$^{-1}$ of free Oxm s.c. (gray) and fibrillar Oxm s.c. (violet). The data correspond to the average measurements from 3 rats ± s.d

fibrillar oligomeric assemblies[39–41]. Nonetheless, peptide oligomerization can also be beneficial to extent the half-life of peptide therapeutics as described for Liraglutide (Victoza)[42]. Potential immunogenicity of protein aggregates is a subject of intense research, and it appears that some self-assembled peptides elicit a wide range of immune response from no detectable to strong antibody responses[43, 44]. Even if the molecular determinants have yet to be established, it has been reported that immunogenicity of self-assembled peptide can be significantly attenuated by modulating the peptide sequence recognized by T cells[43]. Therefore, further studies will be required to fully characterize in vivo toxicity and immunogenicity profiles of fibrillar Oxm.

**Oxm nanofibrils produce a pharmacological effect in vivo.** To understand if fibrillar Oxm can act as a pharmacological depot, we followed glucose levels in wild-type C57BL/6 mice following administration of either 0.09% saline vehicle, free Oxm (10 mg kg$^{-1}$ s.c), or fibrillar Oxm (10 and 20 mg kg$^{-1}$ s.c.). Blood glucose was measured in blood samples taken from tail prick using a hand-held glucometer at either baseline, 45 min, 4 or 6 h post-dose. Area under the curve (AUC) analysis for this 6 h time period showed that both doses of fibrillar Oxm as well as free Oxm had a significant glucose lowering effect vs. vehicle (Fig. 6a). Indeed at 6 h, vehicle blood glucose concentration was 7.8 mM, whereas free Oxm, fibrillar Oxm 10 mg kg$^{-1}$, and fibrillar Oxm 20 mg kg$^{-1}$ were 5.2, 6.3, and 6 mM respectively, all of which were significantly lower than vehicle 6 h after the subcutaneous administration (Fig. 6b). Six hours after administration, an intraperitoneal glucose tolerance test (GTT) was conducted and showed that glucose levels during the test were significantly lower in the fibrillar Oxm group at 20 mg kg$^{-1}$ vs. vehicle at times 45 and 60 min, demonstrating improved glucose tolerance (Fig. 6c). The AUC's calculated during the GTT showed that there was a significant difference vs. vehicle in free and fibrillar groups at both doses (Fig. 6d). The GLP-1 agonist Liraglutide at 0.03 mg kg$^{-1}$ was used as a positive control. These data provide an early insight that nanofibrils of Oxm when dosed in vivo can produce a glucose lowering pharmacodynamic effect, therefore warranting further investigation and optimization as to the therapeutic potential of such formulations.

**Oxm nanofibrils prolong peptide serum bioactivity in vivo.** To investigate the application of fibrillar Oxm as a reservoir from which peptide can be released in vivo, we administered the nanofibrils at 5 and 10 mg kg$^{-1}$ s.c. in CD rats and took serum samples for up to 5 days post-injection. Serum Oxm content was determined as in vitro bioactivity using cell-based hGLP-1R and hGCGR cAMP bioassays, and compared with free Oxm dosed at 5 and 10 mg kg$^{-1}$ s.c. and at 5 mg kg$^{-1}$ i.v. (Fig. 7a, b). At 10 min, Oxm serum concentrations showed an initial peak of 1.34 and 1.85 nM for 5 and 10 mg kg$^{-1}$ of the nanofibrils, respectively, (hGLP-1R cAMP assay), probably due to the presence of free Oxm in the fibrillar sample. This was followed by a steady-state with a constant bioactivity equivalent of 1 nM for 5 days. The hGLP-1R cAMP assay revealed Oxm bioactivity in serum even at 5 days, the final time point we analyzed, at 0.48 nM (after 5 mg kg$^{-1}$) and 1.16 nM (after 10 mg kg$^{-1}$; Fig. 7a). Similarly, using the hGCGR cAMP assay, Oxm concentration in serum was 0.91 and 1.19 nM at 5 days after a single injection of 5 and 10 mg kg$^{-1}$, respectively (Fig. 7b). Overall, with an equivalent of 1 nM Oxm bioactivity detected for 5 days, this study shows a sustained release of Oxm from the nanofibrils in vivo that is characterized by a significantly longer bioactivity than free Oxm. Indeed, Oxm bioactivity is not detected after 2 h post-injection of 5 mg free peptide kg$^{-1}$ (i.v.) or 4 h post-injection of 5 and 10 mg free peptide kg$^{-1}$ (s.c.) (Fig. 7a, b).

The prolongation of Oxm bioactivity is remarkable when considering that no modification has been performed such as engineering poly(ethylene glycol)- and lipid-modified Oxm analogs[4, 24, 26, 45]. The activity of Oxm is limited in vivo by both excretion (i.e., clearance through the kidneys) and enzymatic inactivation through the activity of, for example, dipeptidyl peptidase 4 (DPP4)[46]. Recently, enhanced proteolytic stability was achieved by synthesizing crosslinked Oxm analogs[27]. Even though these analogs showed higher serum stability, their half-life was only prolonged from 0.6 h for the native sequence to 1.9 h for the analogs after s.c. injection.

To further characterize the pharmacokinetic profiles of free and fibrillar Oxm in our rats, a two-compartment model was initially fitted to the i.v. free Oxm hGLP-1R data (5 mg kg$^{-1}$; Supplementary Fig. 4). Pharmacokinetic parameter estimates (Supplementary Table 4) suggest an elimination half-life of ~13 min for free Oxm, similar to what has been reported in humans[25]. Subsequently, hGLP-1R data for free Oxm following s.c. administration (5 and 10 mg kg$^{-1}$) were fitted to an absorption model including two parallel routes of absorption, with and without a delay (lag time; Supplementary Fig. 5). Parameters for the i.v. phase were fixed from the previous analysis of i.v. data. Parameter estimates using the hGLP-1R assay data (Supplementary Table 5) suggested that the bioavailability of free Oxm is ~34% (21% using hGCGR data), and the absorption half-lives of the parallel routes, with and without a lag time, calculated as the ratio of ln (2) and the rate constants, were 3.29 (6.78 using hGCGR assay data) and 1.57 (0.86 using hGCGR assay data) h, respectively. These estimates support the notion that free Oxm is absorbed at a slower rate than it is eliminated.

hGLP-1R data from fibrillar Oxm administration (5 and 10 mg kg$^{-1}$) were fitted to an expanded model that accounts for an additional absorption phase and fibril dissociation, also with two parallel routes with and without a lag time (Supplementary Fig. 6). The possibility of free Oxm being immediately available in the central compartment was also considered, to account for the rapid appearance of a peak of bioactivity in serum (Supplementary Fig. 7). Parameter estimates (Supplementary Table 6) suggest that the absorption phase of fibrillar Oxm was even more rate limiting, with absorption half-lives of up to several days. Given the length of the experiment (5 days), these numerical estimates need to be considered in light of the duration of the experiment, which was shorter than the estimated half-life, but point to a significantly prolonged bioactivity. Interestingly, the data could only be fitted if the central volume of distribution ($V_c$) was

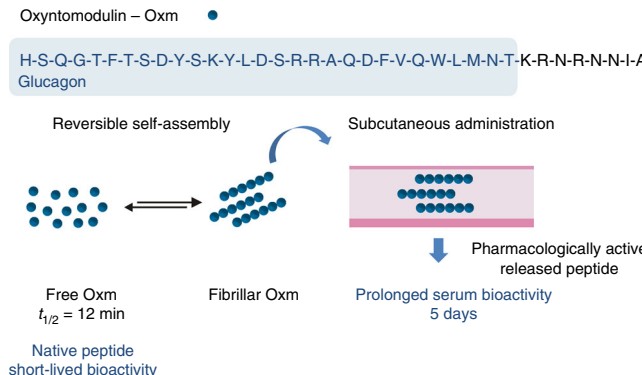

**Fig. 8** Reversible peptide self-assembly to control and prolong Oxm bioactivity in vivo. The proglucagon-derived peptide hormone Oxm has limited clinical application owing to its short half-life of 12 min in humans. Oxm readily self-assembles into nanofibrils which subsequently dissociate under physiological conditions to release pharmacologically active peptide. The subcutaneous administration of the fibrils in rats prolongs peptide serum bioactivity with Oxm detected for at least 5 days post-nanofibril injection compared to free Oxm (which lost all serum bioactivity after only 4 h)

adjusted with respect to the original i.v. estimate. This may reflect a difference in the bioavailability or a change in the molecular properties of Oxm once it is released from the fibrils.

## Discussion

In the present study, we show that peptide hormones such as Oxm, which have a natural propensity to self-assemble, can be formulated as reversibly self-assembling nanofibrils for s.c. administration (Fig. 8). This work describes an approach to develop nanofibril formulations: self-assembly and dissociation under mild conditions, impact on peptide potency, pharmacology, and pharmacokinetics. The 37-amino-acid Oxm peptide readily self-assembles reversibly into nanofibrils, which perform as a slow release depot, without the use of polymeric materials. Fibrillar Oxm delays the absorption rate after s.c. administration, which significantly prolongs peptide presence in serum from a few hours to days, as compared to the free peptide. When tested in mice, the nanofibril formulation significantly lowered blood glucose levels for up to 6 h after administration. After this time a glucose tolerance test showed an improvement in glucose control. This data demonstrates that nanofibril formulation of Oxm can produce a glucose lowering effect similar to free Oxm in normal mice in vivo. The data from these studies provide a platform from which we can build further understanding of the potential benefits of peptide nanofibril formulations, including prolonged PK profiles and efficacy in a range of models. Indeed, we further need to understand the therapeutic potential of such peptide formulations in chronic animal models of diabetes and obesity.

In addition, after being released from the fibrils, Oxm is still subject to rapid renal clearance and proteolytic degradation limiting peptide concentration in serum. Therefore, further improvement of serum exposure and long-acting effects can be achieved by combining self-assembling nanostructures with discrete sequence modifications to increase the serum stability of the released peptide.

Further investigation of the self-assembly process will be carried out but it is likely to occur via a nucleated polymerization reaction followed by events such as fibril elongation and fragmentation as suggested for self-assembling peptides and proteins including the GLP-1 peptide hormone[47, 48]. The equilibrium between fibrillar and monomeric states can be controlled by optimizing self-assembly conditions (e.g., peptide concentration,

pH, presence of salt and excipient) to tune peptide conformation and fibril structure[29, 33] and increase the in vivo performance of the nanofibril formulation.

Finally, this strategy has the potential to be applied to other functional peptides in light of the intrinsic propensity of polypeptide sequences to self-assemble into nanostructures and given that the fibrillar state is exploited by secretory cells to store peptides. For instance, peptide hormones including glucagon, GLP-1, exendin-4, calcitonin, and gastric inhibitory peptide, are known to self-assemble[15], have therapeutic potential but typically suffer from a short half-life. This study opens an area of investigation to assess the clinical application of reversibly self-assembling nanofibrils for potential peptide therapeutics displaying short-lived bioactivity in vivo.

## Methods

**Materials.** Synthetic native human oxyntomodulin (His-Ser-Gln-Gly-Thr-Phe-Thr-Ser-Asp-Tyr-Ser-Lys-Tyr-Leu-Asp-Ser-Arg-Arg-Ala-Gln-Asp-Phe-Val-Gln-Trp-Leu-Met-Asn-Thr-Lys-Arg-Asn-Arg-Asn-Asn-Ile-Ala, acetate salt, MW = 4449.9 g mol$^{-1}$) (≥95% purity) was purchased from Bachem (Switzerland). HPLC grade water (resistivity >18 MΩ cm), PBS (pH 7.4), sodium dodecyl sulfate and human collagen (Bornstein and Traub type I, recombinant, expressed in *Nicotiana*) were purchased from Sigma-Aldrich (UK). Sterile saline (0.9% (w/v) sodium chloride solution) (Baxter) and sterile water for injection BP (Fresenius Kabi) were purchased from VWR (UK). Millex syringe filters (pore size = 0.22 μm), Ultrafree-MC centrifugal filter devices with yellow Durapore membranes (pore size = 0.22 μm) and Millipore Amicon Ultra-0.5 50 K centrifugal filter devices were obtained from Fisher Scientific (UK). Hellmanex III was obtained from Hellma Analytics (Germany). Clear glass vials were obtained from MedImmune Nijmegen manufacturing facility (The Netherlands).

**Preparation of fibrillar oxyntomodulin.** The preparation of Oxm solutions was carried out in a class 2 cabinet using sterile glass vials. Oxm was dissolved at 10–20 mg mL$^{-1}$ in sterile NaOH (4.3 mM) and 0.09% saline solution (diluted from the 0.9% saline solution) to achieve a pH between 7.0 and 7.3. The solution was passed through a 0.22 μm pore size membrane, and peptide concentration was measured before diluting the solution to 10 mg mL$^{-1}$ using 0.09% saline. The solution was then incubated at 37 °C for 5 days with orbital shaking (200 rpm, Thermo Scientific MaxQ 4450 benchtop orbital shaker) and then 2–3 days without agitation. Thereafter, 0.1 mg mL$^{-1}$ of fibrillar Oxm was incubated with 10 mg mL$^{-1}$ free peptide in water for 1 week at 37 °C without agitation. Finally, the 10 mg mL$^{-1}$ fibrillar Oxm solution was diluted to 1 mg mL$^{-1}$ using sterile 0.09% saline, incubated at 37 °C for 2–9 days and then a further 9 days at room temperature before being stored at 4 °C. The conversion yield was assessed by measuring the concentration of the remaining free peptide. The fibrillar material was separated from the free peptide in solution after centrifugation of an aliquot of the solution for 30 min at 16,200×g and filtering the supernatant through a 50 kDa molecular weight cutoff membrane.

**AFM study.** A PicoPlus AFM instrument with a PicoSPM II controller from Molecular Imaging (Agilent) was used for the AFM imaging studies. Images were acquired at room temperature in air using the AC mode with NSC36/no Al cantilevers (Mikromasch, with force constants varying from 0.6 to 2 N m$^{-1}$). For each imaging experiment, an aliquot of the corresponding solutions was deposited onto freshly cleaved mica and left to dry without rinsing.

**Spectroscopic analysis.** Peptide concentrations were measured using a NanoDrop 2000 UV/Vis spectrophotometer (Thermo Scientific, UK). Far-UV circular dichroism spectra (wavelength range: 180–260 nm) were acquired at 22 °C on a Jasco J-815 spectropolarimeter using a 0.1 mm path length cuvette, with a data pitch of 0.5 nm, a 1-nm bandwidth, a scanning speed of 50 nm min$^{-1}$, a 4-s response time, and a 5-scan accumulation. Buffer spectra were measured under the same conditions and subtracted from the sample spectra. Circular dichroism spectra were deconvoluted with the CONTINLL, SELCON3, and CDSSTR algorithms using CDPro software[49, 50]. Fluorescence measurements were carried out on a Varian Cary Eclipse fluorescence spectrophotometer, using a 10 × 2 mm path length cuvette. ThT binding was monitored by exciting the sample at 445 nm and recording the emission fluorescence spectrum from 460 to 600 nm. Freshly prepared and fibrillar Oxm (1 mg mL$^{-1}$) in 0.09% saline were incubated with ThT (50 μM) for 20 min before the measurement. ATR-FT-IR spectra of free and fibrillar Oxm were recorded in a Perkin Elmer Spectrum 400 FT-IR spectrophotometer equipped with a Specac's Golden Gate ATR sampling accessory. A 2-μl sample of free and fibrillar Oxm (1 mg mL$^{-1}$ in 0.09% saline) was placed on top of the ATR accessory and dried with a stream of nitrogen. This procedure was repeated three times. For each samples, 32 interferograms were co-added at 2 cm$^{-1}$ resolution within the range of 1000 to 2000 cm$^{-1}$.

**Dynamic light scattering analysis**. The intensity distribution profiles of freshly prepared and fibrillar Oxm (1 mg mL$^{-1}$) in 0.09% saline at 25 °C were determined by DLS using a Zetasizer Nano ZS (Malvern Instruments, UK) at a back-scattering angle of 173°.

**Stability studies**. Solutions of fibrillar Oxm at 1 mg mL$^{-1}$ in phosphate buffer (25 mM, pH 7.5), Tris-HCl buffer (25 mM, pH 7.5), water (pH 5.9–6.2), 0.09% saline (pH 5.9–6.2) and aqueous HCl (10 mM, pH 2) were incubated for 4 and 48 h under quiescent conditions at 37 °C. The samples were first centrifuged at 16,200×g for 30 min. The collected supernatant was then filtered through a 50 kDa molecular weight cutoff membrane. The concentration of peptide was measured in the filtrate and compared to the initial peptide concentration to assess the percentage release. Released peptides in water were used for further studies. Mass spectra were obtained by matrix-assisted laser desorption ionization (MALDI) on a Bruker ultrafleXtreme mass spectrometer. The net charge of the peptide vs. pH was calculated using GPMAW 9.52a software.

**Dissociation study of fibrillar Oxm using DPI**. A dual polarization interferometer (Farfield Analight 4D, Biolin Scientific AB) was used to optically characterize the dissociation profile of fibrillar Oxm deposited onto collagen-coated sensor chips. Details of the instrumentation has been described previously[51]. Human collagen, diluted to 0.2 mg mL$^{-1}$ in water, was first deposited onto the unmodified oxynitride sensor chip and left to dry at room temperature before being briefly rinsed with water. Then, fibrillar Oxm (1 mg mL$^{-1}$, 10 μL) in 0.09% saline was deposited onto the collagen layer and left to dry at room temperature without rinsing. Water or PBS solutions were continuously flowed over the deposited fibrillar Oxm layer at a rate of 20 μL min$^{-1}$ at 37 °C. After incubation, aqueous 10 mM HCl (pH 2) and 2% Hellmanex in water were injected to remove any residual materials (including the collagen layer) before proceeding to the chip calibration using 80% (w/w) ethanol in water. Data were analyzed using the Analight Explorer 1.6.0.27583 (Farfield-Biolin Scientific AB, Sweden) to calculate the layer refractive index, density, thickness, and mass[51]. Variations in the layer properties were calculated from the maximum values at the start of incubation. As blood is almost phosphate free, the dissociation of fibrillar Oxm was also monitored in a Tris-buffered saline (TBS) and compared to water (Supplementary Fig. 3). Similarly to PBS, slower dissociation was observed in TBS compared to water. Finally, removal of materials from a collagen layer was assessed in water and PBS (Supplementary Fig. 3). A negligible amount of collagen was removed from the surface in water. In PBS, the phase change stabilized after an initial decrease.

**Potency assay**. In vitro agonist potencies of free, released, and fibrillated Oxm were determined in cAMP accumulation assays in CHO cells that were stably transfected with human GLP1R or human GCGR receptor[52–54]. In brief, peptide samples were incubated with cells plated at 500 cells per well in 384-well black shallow microtiter plates (Corning, USA) for 30 min prior to lysis and detection using the HTRF cAMP dynamic 2 assay kit (Cisbio, France). Plates were then incubated for 1 h prior to reading on an Envision plate reader (Perkin Elmer, USA). Eleven-point duplicate concentration-response curves were generated for three independent experiments, and data were represented as the percent activation of the maximum reference ligand. Curves were fitted using nonlinear regression analysis in GraphPad Prism software 6.03 (GraphPad, USA).

**Cell viability assay**. CHO-hGLP-1R cells in growth medium (DMEM, 10%FBS Sigma-Aldrich) were plated at 10,000 cells per well in 96-well black clear-bottom poly-D-lysine-coated microtiter plates (Corning, USA). Cells were pretreated with Oxm or cytotoxic standards (staurosporine, Ro-318220 Sigma-Aldrich) for 48 h, followed by incubation for 5 h with resazurin dye (in vitro toxicological assay kit, Sigma-Aldrich) at 37 °C in a 5% $CO_2$ atmosphere. Fluorescence emission was measured at 590 nm (560 nm excitation).

**Pharmacodynamic studies**. All in vivo procedures were conducted under the authority of a UK Project license which had been reviewed and approved by an Animal Welfare and Ethical Review Body (AWERB) in compliance with EU Directive 2010/63/EU before any work was carried out. Mice were housed under standard conditions with a 12 h light/dark cycle and ad libitum access to food and water. Thirty-nine C57BL/6 wild-type mice were sourced from MRC Harwell, Oxford, UK and randomized into five groups based on body weight (average body weight 33.8 g). Free and fibrillar Oxm were formulated at 1 mg mL$^{-1}$ stock in 0.09% saline. Food was removed from animals at 8 AM, at a time when all animals had a blood glucose measurement made from a drop of blood obtained from a tail prick sample at baseline, 45 min, 2, 4, and 6 h using a hand-held glucometer (Contour NXT, Bayer). Group 1 ($n = 7$) received vehicle 0.09% saline, Group 2 ($n = 8$) received positive control liraglutide 0.03 mg kg$^{-1}$ (4 h later, such that the first injection was 2 h prior to the glucose tolerance test), Group 3 ($n = 8$) received free Oxm at 10 mg kg$^{-1}$ (10 mL kg$^{-1}$), Group 4 ($n = 8$) received fibrillar Oxm 10 mg kg$^{-1}$ (10 mL kg$^{-1}$), and Group 5 ($n = 8$) received fibrillar Oxm 20 mg kg$^{-1}$ (20 mL kg$^{-1}$) via s.c. injection. After 6 h, animals were injected with glucose i.p at 2 g kg$^{-1}$ and glucose measured at 15, 30, 45, 60, 90, and 120 min post-injection. All

data are mean ± s.e.m. Significances were determined using two-sided, unpaired Student's t-test vs. vehicle.

**Pharmacokinetic studies**. All in vivo procedures were conducted under the authority of a UK Project license which had been reviewed and approved by an Animal Welfare and Ethical Review Body (AWERB) in compliance with EU Directive 2010/63/EU before any work was carried out. Lean male CD rats (Charles River Sprague Dawley substrain rats[55]) weighing 200–250 g at the start of the study (Charles River, UK) were housed under standard conditions with a 12 h light/dark cycle and ad libitum access to food and water. Free and fibrillar Oxm were formulated at 1 mg mL$^{-1}$ stock in 0.09% saline. Groups of three animals were injected with either free Oxm at 5 mg kg$^{-1}$ i.v. (5 mL kg$^{-1}$), free Oxm at 5 or 10 mg kg$^{-1}$ s.c. (5 or 10 mL kg$^{-1}$) or fibrillar Oxm at 5 or 10 mg kg$^{-1}$ s.c. (5 or 10 mL kg$^{-1}$). Serum samples were collected from i.v.-injected animals at 5, 10, 15, 20, 30, 45 min and 1, 2, 4, 6, and 24 h and from s.c.-injected animals at 5, 10, 15, 30, 45 min, 1, 2, 4, 6, and 24 h and 2, 3, and 5 days. Blood samples were taken into serum collecting tubes containing DPP4 inhibitor (Millipore, UK), allowed to clot for 30 min and then spun to obtain serum, which was stored at −80 °C until the assays were performed. Serum peptide content was determined as apparent concentration measured by ex vivo bioactivity in serum using an in vitro cell-based cAMP bioassay (Cisbio, France) for determining agonist bioactivity at human GLP1R and human glucagon receptor as described in potency assay above[21, 56]. In brief, CHO K1 cells stably transfected with either human GLP1R or human glucagon receptor were used to determine peptide concentrations in serum by comparing the degree of cAMP accumulation in serum samples from treated animals against a standard curve generated by spiking peptide into naïve pooled male rat serum (Seralabs, West Sussex, UK) using acoustic dispensing. Bioactivity data were analyzed using nonlinear regression analysis in GraphPad Prism. SAAM II software (The Epsilon Group, Charlottesville, VA; version 2.1) was used for the pharmacokinetic analysis. Naive pooling and model-based weights were used for nonlinear regression. The final model is shown in Supplementary Fig. 6.

**Data availability**. All data supporting the findings of this study are available within the article and Supplementary Information file, or available from the authors upon request.

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

## Acknowledgements

We thank the Department of Biochemistry of the University of Cambridge for the use of the Protein and Nucleic Acid Chemistry Facility, which provided MALDI mass analysis service. We thank Dr. Jiro Matsuo and Dr. Marcus Swann for providing polished DPI chips. We would like to thank Helen Brant for help with the in vivo studies. This work was supported by MedImmune Ltd.

## Author contributions

M.M.O., A.L.G.D.S. and M.E.W. conceived the project. M.M.O., A.L.G.D.S., D.C.H., D.B., D.J.C. and P.V. conceived and designed the experiments. M.M.O., S.K., S.M., D.C.H., D.B., J.N. and L.S. performed the experiments. M.M.O., A.L.G.D.S., S.M., D.C.H., D.B., J.N., D.J.C., J.H. and P.V. Analyzed the data. M.M.O., A.L.G.D.S., S.K., S.M., D.C.H., D.B., D.J.C., J.H., P.V. and M.E.W. wrote the paper. All authors discussed the results and commented on the manuscript.

## Additional information

**Competing interests:** The authors declare no competing financial interests.

