## [Peer Review File · Nature Communications]

Reviewers' comments:

Reviewer #1 (Remarks to the Author):

Welland et al show that the natural peptide oxyntomodulin being a potential drug in the treatment of obesity once aggregated into amyloids is showing a long duration of action after subcutaneous delivery attributed to the slow continuous release of soluble functional peptide. This is a very well written, well done, beautiful and very important study on the use of hormone amyloids in drug delivery.

The following points should be considered by the authors

Major points:

(i) The authors consider PBS buffer as the physiological relevant condition for their study. This is however not correct since blood is almost phosphate free. A recent study by Nespovityaya et al (2016) showed that the presence of phosphate interferes strongly with the dissociation of hormone amyloid fibrils, attributed to the Arg/Lys-rich sequence of their hormone and their repulsion reduction by the polyanionic phosphate. Also the present hormone is Arg and Lys-rich and it is thus not surprising that the oxyntomodulin fibrils in the present study dissociate faster in water when compared with PBS buffer. It is suggested to explain the different stability of the oxyntomodulin fibrils in water and phosphate under this aspect.

(ii) Because of the sticky nature of amyloids, the use of fibrillar material may cause large sample-dependent differences in the accuracy of the amount of hormone peptide delivered. This may cause a significant health issue in particular in a non life-threatening disease as obesity (short term). It is thus requested to show the reproducibility level on the amount of hormones after taking defined amounts of de novo prepared hormone amyloids generated by different starting batches.

Minor points:

(i) The authors state that "the conversion yield of the self-assembly of Oxm into fibrillar nanostructures was estimated to be 99% under this condition". It is requested to explain in short the method of measurement.

(ii) The authors state "By contrast, in water 37% of the fibrillar peptide was released after 4 h incubation, and 53% was released by 48 h (Supplementary Table 1). Again it is requested to mention the method of measurement.

Reviewer #2 (Remarks to the Author):

This well-written paper tackles a difficulty in application of bioactive peptides as drugs due to their poor bioavailability. The approach uses peptide's self-assembly property to form amyloid-like fibrils as a depot, creating a controlled-release formulation. The utilization of amyloids as stable depots for long-acting drug has been suggested before and tested on analogs of gonadotropin-releasing hormone.

The specific peptide investigated here is oxyntomodulin, a 37-amino acid proglucagon-derived peptide hormone with sequence homology to both glucagon and glucagon-like peptide-1 (GLP-1), suggested as a potential treatment for diabetes and obesity.

The results presented show that oxyntomodulin fibrillates in a revisable process, to release biologically active peptides. Using the fibrillary formulation, s.c. administration resulted in prolonged presence of oxyntomodulin in rat serum (few days vs. few hours in the free form) at an approximated concentration that is potentially above the pharmacologically effective concentration. The authors used a new surface-based technique, dual polarisation interferometry

(DPI), to assess the real-time dissociation profile of the fibrils in conditions theoretically mimicking s.c. compartments.

Specific comments:

1. Amyloid characterization is not straightforward. Conversion to beta-rich species can be difficult to detect, especially since there is a transition from solution to solid phase (see a note regarding the deconvolution of the CD spectra). Theoretically, information about the fibril themselves is needed (e.g., fibril diffraction, FTIR, solid-phase CD).

At any case, I am not sure about the significance of defining oxyntomodulin fibrils as amyloids, as the author are not attempting to define a new amyloid, but rather describe the utilization of the fibrils as depot for a controlled-release formulation of a therapeutic peptide. The main question related to amyloids is their potential risk as aggregators of human proteins, inducing aggregation diseases (as reported for contaminated human growth hormone from cadavers administered during the 80's). The authors addressed the potential problem of immunogenicity, but other risks should be taken into consideration, especially for none life-saving therapy, given at relatively young ages. Altogether, amyloid-based therapy will have to be examined over long-term studies. In the short term, potential cross-seeding with known disease-associated amyloid culprits can be assessed.

2. ThT fibrillation kinetics should be assessed in order to observe the standard nucleation and aggregation phases. Maybe even the steady release of the peptides from the fibrils could be observed with ThT kinetics.

3. I expect difficulties in controlling the exact dosage from such fibrillary formulation. Amyloid polymorphism and large variability in fibrillation kinetics would affect the preparation of reproducible, consistent, formulations.

4. Deconvolution of the CD spectra: CD spectroscopy measures the solution phase (unless solid-state CD is used). Correspondingly, the spectrum and its deconvolution depend greatly on the input concentration. It needs to be taken into account that during aggregation, the effective concentration of the soluble peptides decreases significantly. A way to overcome this difficulty is a real-time measurement of the protein concentration using UV absorbance taken at the same time as the CD measurements by the same instrument. This will give more accurate estimation of the secondary structure elements and the transition into fibrils. This is not (yet) the standard in amyloid research, but should be.

- Can the author comment about the smoothing of the spectra in fig. 2d & 3a?
- Also, I believe that the software mentioned for the deconvolution should be referenced.
- Figure S1 should also show the spectra.

5. Is the EC50 determined in CHO cells relevant to the effective therapeutic concentration in humans? It is important to provide physiologically accurate numbers here in order to assess whether the estimated serum levels of 1nM released from the fibrils are relevant. Also, is there a more direct method to measure serum levels of the oxyntomodulin?

6. Standard deviations reported in the potency assay were somewhat confusing for me. I am guessing that they represent confidence intervals?

7. Define and referenced "CD rats". Why were these particular rats chosen? Was there any reason not to assess the rats' weight following the oxyntomodulin treatment? I am guessing that there are also some diabetic parameters that could have been assessed after a short treatment. Was there any particular reason for not testing the actual therapeutic effects in the rats?

8. I'm missing methodology for the estimation of pharmacokinetic parameter. How relevant is the model considering that peptide levels were not measured directly?

Reviewer #3 (Remarks to the Author):

The manuscript by Dr. Ouberai and colleagues shows that oxyntomodulin (Oxm) self-assembles into unstable nanofibril formulation which subsequently dissociates under physiological conditions

to release intact and active peptide. Administration of the nanofibrils in rats resulted in prolonged circulating bioactive oxyntomodulin compared to the administration of native oxyntomodulin.

The data are interesting. This reviewer has a few questions.

For a therapeutic approach:

- What is the projected exposure for efficacy in vivo? What is the indication? Weight loss, glucose lowering, etc? The authors should provide at least efficacy during a glucose tolerance test and weight loss.
- What is the vehicle used for the in vivo studies? Please add a comment on vehicle, device and frequency of injection considered for this approach. Is this a potentially viable based on preliminary data in vivo (COG)?

- Please describe the plan to characterize the equilibrium properties of the system under various conditions to address the instability.

- Please comment on the immunogenicity risk of the aggregates. Was any evaluation in rodents and/or higher species performed? How are the authors planning to address this risk in humans?

- Please discuss the different subcutaneous space and preclinical species selection to predict the PK profile in humans.

Thank you

Response to reviewers

Reviewer #1 (Remarks to the Author):

Welland et al show that the natural peptide oxyntomodulin being a potential drug in the treatment of obesity once aggregated into amyloids is showing a long duration of action after subcutaneous delivery attributed to the slow continuous release of soluble functional peptide. This is a very well written, well done, beautiful and very important study on the use of hormone amyloids in drug delivery.

The following points should be considered by the authors

Major points:

(i) The authors consider PBS buffer as the physiological relevant condition for their study. This is however not correct since blood is almost phosphate free. A recent study by Nespovitaya et al (2016) showed that the presence of phosphate interferes strongly with the dissociation of hormone amyloid fibrils, attributed to the Arg/Lys-rich sequence of their hormone and their repulsion reduction by the polyanionic phosphate. Also the present hormone is Arg and Lys-rich and it is thus not surprising that the oxyntomodulin fibrils in the present study dissociate faster in water when compared with PBS buffer. It is suggested to explain the different stability of the oxyntomodulin fibrils in water and phosphate under this aspect.

Reply: We thank the reviewer for this comment; we agree that blood is almost phosphate free. We have now included experimental data describing the stability of fibrillar Oxm in Tris-HCl and phosphate buffers that show no significant difference in the stability of fibrils between these two buffers (see Results section page 7 and Supplementary Table 1). The dissociation profile of fibrillar Oxm was also assessed with DPI in Tris-buffered saline for which, similarly to PBS, slower dissociation was observed compared to water (added to the Methods section page 19 and in Supplementary Figure S3). We have commented on the different stability in the Results section pages 7 and 8.

We would like to point out that the objective of the *in vitro* testing of fibril dissociation in PBS (using DPI) is to characterize the dissociation process under conditions mimicking physiological pH, ionic strength and temperature, with the fibrils in interaction with one main component of the s.c. space, and with peptide cleared upon release from the fibrils. This has been now clarified in the Results section page 8. Currently, there is no regulatory standard for *in vitro* release testing of nano-sized dosage forms. *In vivo-in vitro* correlation is difficult to achieve and we are not trying to reproduce the s.c. space *in vitro* due to its high complexity. Even if we appreciate the importance of salts in the dissociation profile of fibrils, this is not the scope of this work and this will be assessed in a subsequent study.

(ii) Because of the sticky nature of amyloids, the use of fibrillar material may cause large sample-dependent differences in the accuracy of the amount of hormone peptide delivered. This may cause a significant health issue in particular in a non life-threatening disease as obesity (short

term). It is thus requested to show the reproducibility level on the amount of hormones after taking defined amounts of de novo prepared hormone amyloids generated by different starting batches.

Reply: We appreciate the fact that some amyloid fibrils are known to be sticky, but we don't think this statement can be generalized to any kind of amyloid-like fibrils as it also depends on the peptide/protein sequence and solution conditions. In addition, stickiness of biologics or any kind of polymeric materials (used for subcutaneous depot) is also a problem that is mitigated via optimization of the formulation conditions. In addition, dose accuracy also depends on the final device used for administration. Further optimization of the fibril formulation properties related to the dosing such as viscosity will be outperformed in a subsequent study.

Minor points:

(i) *The authors state that "the conversion yield of the self-assembly of Oxm into fibrillar nanostructures was estimated to be 99% under this condition". It is requested to explain in short the method of measurement.*

Reply: We do explain the method of measurement in the Methods section:

“The conversion yield was assessed by measuring the concentration of the remaining free peptide after centrifugation of an aliquot of the solution for 30 min at 16 200 x g and filtering the supernatant through a 50 kDa molecular weight cut-off membrane.” We used UV/Vis spectrophotometry for measuring the concentration of peptide as explained in the Methods section: “Peptide concentrations were measured using a NanoDrop 2000 UV/Vis spectrophotometer (Thermo Scientific, UK).”

In order to clarify the method we have now added in the Methods section pages 16 and 17: “The conversion yield was assessed by measuring the concentration of the remaining free peptide. The fibrillar material was separated from the free peptide in solution after centrifugation of an aliquot of the solution for 30 min at 16 200 x g and filtering the supernatant through a 50 kDa molecular weight cut-off membrane.”

(ii) *The authors state "By contrast, in water 37% of the fibrillar peptide was released after 4 h incubation, and 53% was released by 48 h (Supplementary Table 1). Again it is requested to mention the method of measurement.*

Reply: We do explain the method of measurement in the Methods section page 18:

“Solutions of fibrillar Oxm at 1 mg/mL in phosphate buffer (25 mM pH 7.5), Tris-HCl buffer (25 mM pH 7.5), water, 0.09% saline and 10 mM HCl (pH 2) were incubated for 4 h and 48 h under quiescent conditions at 37°C. The samples were first centrifuged at 16 200 x g for 30 min. The collected supernatant was then filtered through a 50 kDa molecular weight cut-off membrane. The concentration of peptide was measured in the filtrate and compared to the initial peptide concentration to assess the percentage release.”

Reviewer #2 (Remarks to the Author):

This well-written paper tackles a difficulty in application of bioactive peptides as drugs due to

their poor bioavailability. The approach uses peptide's self-assembly property to form amyloid-like fibrils as a depot, creating a controlled-release formulation. The utilization of amyloids as stable depots for long-acting drug has been suggested before and tested on analogs of gonadotropin-releasing hormone.

The specific peptide investigated here is oxyntomodulin, a 37-amino acid proglucagon-derived peptide hormone with sequence homology to both glucagon and glucagon-like peptide-1 (GLP-1), suggested as a potential treatment for diabetes and obesity.

The results presented show that oxyntomodulin fibrillates in a reversible process, to release biologically active peptides. Using the fibrillary formulation, s.c. administration resulted in prolonged presence of oxyntomodulin in rat serum (few days vs. few hours in the free form) at an approximated concentration that is potentially above the pharmacologically effective concentration. The authors used a new surface-based technique, dual polarisation interferometry (DPI), to assess the real-time dissociation profile of the fibrils in conditions theoretically mimicking s.c. compartments.

Specific comments:

1. Amyloid characterization is not straightforward. Conversion to beta-rich species can be difficult to detect, especially since there is a transition from solution to solid phase (see a note regarding the deconvolution of the CD spectra). Theoretically, information about the fibril themselves is needed (e.g., fibril diffraction, FTIR, solid-phase CD).

Reply: We appreciate the fact that amyloid fibril characterization is not straightforward. However, we believe that we have provided strong evidence that oxyntomodulin self-assembles into fibrillar structures displaying the generic features of amyloid-like fibrils: fibrillar structures of approximately 10 nm in diameter, binding to the cross-beta sheet probe Thioflavin T, change in conformation to beta-rich fibrillar structures. As requested, we have now added ATR-FTIR data in the Results section page 6 and in Figure 2 to support further the structural properties of the fibrils and to show a structural transition to β -sheet structure.

At any case, I am not sure about the significance of defining oxyntomodulin fibrils as amyloids, as the author are not attempting to define a new amyloid, but rather describe the utilization of the fibrils as depot for a controlled-release formulation of a therapeutic peptide. The main question related to amyloids is their potential risk as aggregators of human proteins, inducing aggregation diseases (as reported for contaminated human growth hormone from cadavers administered during the 80's). The authors addressed the potential problem of immunogenicity, but other risks should be taken into consideration, especially for none life-saving therapy, given at relatively young ages. Altogether, amyloid-based therapy will have to be examined over long-term studies. In the short term, potential cross-seeding with known disease-associated amyloid culprits can be assessed.

Reply: We thank the reviewer for this comment. We agree that this potential risk has to be further investigated. This is the subject of some ongoing work. However, we believe that this risk is minimal as disease-related proteins or peptides forming these aggregates (for instance beta-amyloid peptide, alpha-synuclein and prion) have very different sequences. Indeed, it has been reported that cross-seeding is promoted by sequence similarity and conformational compatibility

(for example “Krebs, M. R. H. et al. Observation of sequence specificity in the seeding of protein amyloid fibrils, *Protein Science*, 13, 1933–1938 (2004)).

2. ThT fibrillation kinetics should be assessed in order to observe the standard nucleation and aggregation phases. Maybe even the steady release of the peptides from the fibrils could be observed with ThT kinetics.

Reply: The focus of this work is to demonstrate the use of fibrillar structures made of a native and bioactive peptide to prolong its bioactivity in serum. As widely described for amyloid fibril formation (for example in the review: Knowles, T. P. J., Vendruscolo, M. & Dobson, C. M. The amyloid state and its association with protein misfolding diseases. *Nat. Rev. Mol. Cell Biol.* 15, 384–96 (2014)), it is likely that the process occurs via a nucleated polymerization reaction followed by events such as fibril elongation and fragmentation. We appreciate the high importance of understanding the mechanism of oxyntomodulin fibril formation, but we believe that this process requires a detailed characterization. Therefore, a full description of the kinetics and thermodynamics of this system will follow in a subsequent study. We have now commented this in the Discussion section page 14.

3. I expect difficulties in controlling the exact dosage from such fibrillary formulation. Amyloid polymorphism and large variability in fibrillation kinetics would affect the preparation of reproducible, consistent, formulations.

Reply: We appreciate the fact that fibril polymorphism can affect the nanofibril formulation. However, we are controlling the preparation process of the fibrillar material by using a seeding method which consists of adding preformed fibrils to a solution of fresh peptide. This method promotes the elongation process and bypasses the primary nucleation process which is causing variability in the kinetics. In addition, reports on glucagon show that by controlling peptide concentration, homogeneous batches of straight or twisted fibrils can be obtained (Andersen, C. B. et al. Glucagon fibril polymorphism reflects differences in protofilament backbone structure. *J. Mol. Biol.* 397, 932–946 (2010)). We aim, in the future, to fully understand the factors affecting the structural properties and kinetics of fibrillar Oxyntomodulin to optimize the homogeneity of the fibril batches and the formulation properties.

4. Deconvolution of the CD spectra: CD spectroscopy measures the solution phase (unless solid-state CD is used). Correspondingly, the spectrum and its deconvolution depend greatly on the input concentration. It needs to be taken into account that during aggregation, the effective concentration of the soluble peptides decreases significantly. A way to overcome this difficulty is a real-time measurement of the protein concentration using UV absorbance taken at the same time as the CD measurements by the same instrument. This will give more accurate estimation of the secondary structure elements and the transition into fibrils. This is not (yet) the standard in amyloid research, but should be.

Reply: We thank the reviewer for this comment and agree that CD measures the solution phase. However, as also reported for the self-assembly of many peptides and proteins including glucagon, CD spectroscopy is a standard technique to show a change in the secondary structure of the peptide from helical to beta-sheet contents upon fibril formation (Ghodke, S. et al.

Mapping out the multistage fibrillation of glucagon. FEBS J. 279, 752–765 (2012)). ATR-FTIR data have now been included in the Results section page 6 and in Figure 2e of the manuscript supporting a transition from helical to beta-sheet content. The deconvolution of the CD spectra previously shown in Figure 2e has been moved to the supplementary information and both CD and FTIR spectra (Figure 2d and 2e) support the structural transition into fibrils.

• *Can the author comment about the smoothing of the spectra in fig. 2d & 3a?*

Reply: We didn't smooth the data presented in the spectra figures 2d and 3a. The spectra were measured in the range of 260 to 180 nm with a data pitch of 0.5 nm, a 1-nm bandwidth, a scanning speed of 50 nm/min, a 4-s response time, and a 5-scan accumulation. Buffer spectra were measured under the same conditions and subtracted from the sample spectra. This has been added to the Methods section page 17.

• *Also, I believe that the software mentioned for the deconvolution should be referenced.*

Reply: We have now referenced the deconvolution in the Methods section page 17: "Circular dichroism spectra were deconvoluted with the CONTINLL, SELCON3 and CDSSTR algorithms using CDPro software^{47,48}."

• *Figure S1 should also show the spectra.*

Reply: The spectra is shown in Figure 3a as mentioned in the Results section page 7: "Furthermore, the far-UV spectrum of released Oxm in water shows that the peptide recovered its initial conformation, which is characterised by helical and disordered structures in proportions similar to free peptides (Fig. 3a and Supplementary Fig. S2)."

5. *Is the EC50 determined in CHO cells relevant to the effective therapeutic concentration in humans? It is important to provide physiologically accurate numbers here in order to assess whether the estimated serum levels of InM released from the fibrils are relevant.*

Reply: EC50 determined in CHO cells are used for a number of reasons within the manuscript. Initially EC50s in CHO cells expressing either human GLP-1 or Glucagon receptors are shown to demonstrate relative weak/inactive potency of fibrils to free oxyntomodulin and that full potency is recovered in released oxyntomodulin.

Active molecules with potencies determined in these assays have shown good translation to activity in endogenous receptor physiologically relevant cell line assays (such as beta cell lines and hepatocytes) and this has translated to efficacious molecules in pre-clinical rodent and NHP models (Henderson, S. J. et al. Robust anti-obesity and metabolic effects of a dual GLP-1/glucagon receptor peptide agonist in rodents and non-human primates. Diabetes Obes. Metab. 18, 1176–1190 (2016)).

To demonstrate further the relevance of the EC50s and peptide serum levels determined using the bioassay, a pharmacodynamic study has been added to the manuscript showing that s.c. administration of fibrillar Oxm produces a significant glucose lowering effect in mice (as described in the Results section pages 10 and 11, additional Figure 5 and Discussion section page 13, 14 and 15). At the doses used in this study we have shown a pharmacological effect on glucose lowering suggesting that the amount of oxyntomodulin released in serum is sufficient.

Strategies are being investigated to maximize further this pharmacological effect which will influence the human dose prediction.

Also, is there a more direct method to measure serum levels of the oxyntomodulin?

Reply: Reports have shown the lack of available sensitive and specific methods for reliable *in vivo* detection of oxyntomodulin (for example: Bak, M. J. Specificity and sensitivity of commercially available assays for glucagon and oxyntomodulin measurement in humans, *Eur. J. Endocrinol.* 170, 529-538 (2014)).

We believe that our validated assays provide reliable oxyntomodulin apparent concentration or *ex-vivo* bioactivity detected in serum; CHO- receptor cAMP accumulation assays are used as bioactivity assay to determine the apparent concentration of peptide in serum samples from rodent studies by reading from a serum standard curve spiked with free oxyntomodulin or released oxyntomodulin. This method has been reported previously for oxyntomodulin and other glucagon GLP1 dual agonist molecules in a number of papers.

References of papers using CHO-hGLP-1R assay have been added to the Methods section of the manuscript page 21: Pocai A. et al. Glucagon-Like Peptide 1/Glucagon Receptor Dual Agonism Reverses Obesity in Mice. *Diabetes* 58, 2258-2266 (2009); Kosinski, J. R. et al. The Glucagon Receptor Is Involved in Mediating the Body Weight-Lowering Effects of Oxyntomodulin. *Obesity* 20, 1566–1571 (2012).

6. Standard deviations reported in the potency assay were somewhat confusing for me. I am guessing that they represent confidence intervals?

Reply: We have now updated the Supporting table 3 with additional EC50 data to report geometric mean and standard error of the mean from 5-7 independent experiments.

7. Define and referenced "CD rats". Why were these particular rats chosen? Was there any reason not to assess the rats' weight following the oxyntomodulin treatment? I am guessing that there are also some diabetic parameters that could have been assessed after a short treatment. Was there any particular reason for not testing the actual therapeutic effects in the rats?

Reply: CD rats are very common normal lean Sprague Dawley strain of rats produced by Charles River since the 1950s. CD rats have been defined and referenced in the Methods section page 21. We use these for our in-house rat PK studies for all projects and they are not a model of metabolic diseases. PK studies are typically run as n=3 with no vehicle group and furthermore are not powered to study metabolic phenotype such as body weight and glucose control. The PK study aims to measure Oxyntomodulin bioactivity in serum and compare the extent of this bioactivity between fibrillar Oxm administrated subcutaneously and free Oxm administrated subcutaneously and in IV.

We have now added pharmacodynamic data collected in a glucose tolerance test study in mice showing that the s.c. administration of the nanofibril formulation produces glucose lowering as described in the Results section pages 10 and 11, additional Figure 5 and Discussion section page 13, 14 and 15. This was a single dose acute study to assess glucose lowering, further chronic and repeat dose study would need to be conducted in order to elucidate a potential body weight effect.

8. *I'm missing methodology for the estimation of pharmacokinetic parameter. How relevant is the model considering that peptide levels were not measured directly?*

Reply: We believe that the methodology for the estimation of pharmacokinetic parameter is described in sufficient detail in the Results and Methods sections pages 11, 12 and 21 and in the supplementary information. We have described the relevance of the peptide levels detected in serum in the reply to question 5. We have now further clarified serum peptide levels in the Methods section of the manuscript page 21: "Serum peptide content was determined as apparent concentration measured by *ex-vivo* bioactivity in serum using an *in vitro* cell-based cAMP bioassay (Cisbio, France) for determining agonist bioactivity at human GLP1R and human glucagon receptor as described in potency assay above^{21,56}. ...Bioactivity data were analysed using nonlinear regression analysis in GraphPad Prism. SAAM II software (The Epsilon Group, Charlottesville, VA; version 2.1) was used for the pharmacokinetic analysis."

Reviewer #3 (Remarks to the Author):

The manuscript by Dr. Ouberai and colleagues shows that oxyntomodulin (Oxm) self-assembles into unstable nanofibril formulation which subsequently dissociates under physiological conditions to release intact and active peptide. Administration of the nanofibrils in rats resulted in prolonged circulating bioactive oxyntomodulin compared to the administration of native oxyntomodulin.

The data are interesting. This reviewer has a few questions.

For a therapeutic approach:

- What is the projected exposure for efficacy in vivo? What is the indication? Weight loss, glucose lowering, etc? The authors should provide at least efficacy during a glucose tolerance test and weight loss.

Reply: As requested, we have now added efficacy data collected in a glucose tolerance test *in vivo* showing that s.c. administration of the nanofibril formulation produces glucose lowering in mice (Results section pages 10 and 11, additional Figure 5 and Discussion section page 13, 14 and 15). It is clear that the doses used in this study are sufficient to see significant glucose lowering effect in mice. This was a single dose acute study to assess glucose lowering, further chronic and repeat dose study would need to be conducted in order to elucidate a potential body weight effect. Strategies are also being investigated to maximize further this pharmacological effect as commented in the Discussion section pages 13, 14 and 15.

We would like to stress that the main objective of this work is to apply supramolecular self-assembly to a native and biologically active peptide, known to have a therapeutic effect and used here as a model system, to improve its pharmacokinetics. We believe that this work clearly shows a pharmacokinetic difference between free and fibrillar peptide while maintaining efficacy.

- What is the vehicle used for the in vivo studies? Please add a comment on vehicle, device and frequency of injection considered for this approach. Is this a potentially viable based on preliminary data in vivo (COG)?

Reply: The *in vivo* study that was described in the first version of this manuscript is a pharmacokinetic study which doesn't require any vehicle group as we are not assessing any therapeutic effect. The pharmacokinetic study aims to measure oxyntomodulin bioactivity in serum and compare the extent of this bioactivity between fibrillar Oxm administrated subcutaneously and free Oxm administrated subcutaneously and IV.

For the pharmacodynamic study performed in mice and now added to this manuscript, 0.09% saline was used as a vehicle.

We believe that the points raised about device, frequency of injection, cost of goods (COG) are out of scope as we are not yet proposing to make this into a product. We are simply using oxyntomodulin to exemplify that making nanofibrils from peptide is a beneficial strategy to reduce dose frequency while maintaining efficacy. We will determine the appropriate device and frequency of injection, and assess COG for subsequent projects which will use this approach, but they will also depend on the indication and the target therapeutic exposure. However, we can anticipate that the benefit to reduce dose frequency with a depot of peptides released in a constant rate will reduce the burden on patient and reduce the cost of consumable. This is potentially viable based on *in vivo* studies showing that the s.c. administration of the nanofibril formulation produces glucose lowering and significantly prolongs peptide presence in serum from a few hours to days, as compared to the free peptide.

- *Please describe the plan to characterize the equilibrium properties of the system under various conditions to address the instability.*

Reply: The plan to characterize the equilibrium properties of the system is first based on the stability and dissociation studies as described in the Results section of the manuscript pages 7, 8 and 9. We take two approaches: one looking at the stability of the fibrils at a given concentration and at equilibrium under various solution conditions. We measured significant differences on the stability of the fibrils by assessing the effect of a variation in pH and ionic strength. The fibrils are stable at equilibrium with a small amount of NaCl or in presence of phosphate and Tris-HCl buffers but are less stable without salt and in an acidic pH. In the second approach, we show that when the equilibrium is shifted by clearing released peptide (DPI experiment) the fibrils dissociate at a physiological pH and in presence of a large amount of salts (PBS, TBS).

In a subsequent study (in preparation), we are characterizing the thermodynamics and kinetics of the fibrils by changing peptide concentration, ionic strength, temperature etc. This is explained in the manuscript in the Results section pages 8 and 9.

- *Please comment on the immunogenicity risk of the aggregates. Was any evaluation in rodents and/or higher species performed? How are the authors planning to address this risk in humans?*

Reply: We agree with the reviewer on the immunogenicity risk of the fibrils like any other biologics or drug delivery materials. We have not performed any evaluation in rodents and/or higher species yet but plan to assess this risk. We have commented on this in the Results section of the manuscript page 10. "Potential immunogenicity of protein aggregates is a subject of intense research, and it appears that some self-assembled peptides elicit a wide range of immune response from no detectable to strong antibody responses^{43,44}. Even if the molecular determinants have yet to be established, it has been reported that immunogenicity of self-assembled peptide can be significantly attenuated by modulating the peptide sequence recognized by T cells⁴³."

Therefore, further studies will be required to fully characterize in vivo toxicity and immunogenicity profiles of Fibrillar Oxm.”

- Please discuss the different subcutaneous space and preclinical species selection to predict the PK profile in humans.

Reply: Regarding the different subcutaneous space, we think it is difficult to indicate a physiological correlate with the modelled subcutaneous depot spaces. The subcutaneous space consisted of two separate compartments; fibrillar and free oxyntomodulin depots, but their complexity is largely empirically driven by the shape of the curve and represents the rate(s) at which the molecule is absorbed; thus, it would be difficult at this point, from the limited data, to indicate which physical spaces these refer to. Regarding the second part of the question, we think human PK prediction is outside the scope of this manuscript. At this stage of the project we have not performed any formal allometric predictions of human PK, as we aimed to first assess a PK difference between formulations. In addition, rodent s.c. PK is not sufficient for proper allometric scaling of peptide molecules to man, primate PK data would be required.

Reviewers' comments:

Reviewer #1 (Remarks to the Author):

The revised version of the manuscript by Ouberai et al on the controlled release of peptide hormone from amyloids is improved (and still of highest importance and relevance). However, still several questions remain:

(i) It is mechanistically unclear for the reviewer why the amyloid dissociate in water when compared to Tris-HCL and PBS (new Table S2). The reviewer asks to indicate the pH of the amyloid sample in water and if this is low, the concept of a dilution effect upon release in blood may be of interest to discuss.

(ii) The problem of reproducibility of peptide release from the amyloids, the potential presence of polymorphs (see also reviewer 2 remarks), and the administration of the same dose of potentially sticky amyloids have not been resolved adequately and it is thus of concern. This issue is also conceptually important, since the approach undertaken in the published and references GNRH analog studies started mainly with soluble material that aggregated in the body, while the presented approach incubates amyloids. The reviewer requests therefore again to measure in vitro the reproducibility of the release of hormones after taking defined amounts of de novo prepared hormone amyloids generated by different starting batches. In this context, the Table S1 is cited, which shows actually such data, but it is unclear whether different batches and starting material was used or whether the data come from a single experimental set up.

Reviewer #2 (Remarks to the Author):

In the revised manuscript, the authors added requested information and addressed feasible major issues.

I have one comment to the authors regarding the risks of amyloid-based therapy: I don't agree with the response that "we believe that this risk is minimal as disease-related proteins or peptides forming these aggregates (for instance beta-amyloid peptide, alpha-synuclein and prion) have very different sequences...". Although some publications suggested that sequence similarity is required, many studies showed cross-seeding between remote systems. In any case, I don't think that this is a major issue to be addressed here since this manuscript presents a proof of concept and not an actual drug. The studies of amyloid-based therapy will require many years in order to assess the actual risks.

Reviewer #3 (Remarks to the Author):

The authors have addressed all the issues raised

Response to reviewers

Reviewer #1 (Remarks to the Author):

The revised version of the manuscript by Ouberaï et al on the controlled release of peptide hormone from amyloids is improved (and still of highest importance and relevance).

Reply: We thank the reviewer for this comment.

However, still several questions remain:

(i) It is mechanistically unclear for the reviewer why the amyloid dissociate in water when compared to Tris-HCL and PBS (new Table S2). The reviewer asks to indicate the pH of the amyloid sample in water and if this is low, the concept of a dilution effect upon release in blood may be of interest to discuss.

Reply: The pH of the fibril sample in water is between 5.9 and 6.2, similar to the pH measured for the fibril sample in 0.09% saline (now indicated in the Methods section). In order to further assess the effect of salt versus pH we have measured the stability of the fibrils in 25 mM phosphate pH 6 and didn't detect any released peptide after 4h incubation (not included in the manuscript as the mechanistic description of the effect of salts and pH on fibril dissociation is not the scope of this work). The stability observed at pH 6 in 25 mM phosphate is similar to the one observed in phosphate and Tris buffers pH 7.5.

Therefore, as already commented in the manuscript: "The presence of salts such as NaCl, phosphate or Tris-HCl stabilizes the fibrillar state at 1 mg/mL. We therefore anticipate that different equilibrium states are reached depending on the solution conditions, such as the peptide concentration, the presence of various salts and pH. Investigations to further characterise the equilibrium properties of the system under various conditions are ongoing. "

This first *in vitro* experimental setup is assessing the stability of the fibrils at a given concentration and at equilibrium under various solution conditions. In the second *in vitro* experimental setup used to assess fibril stability (using DPI), we show that when the equilibrium is shifted by clearing released peptide, fibrils dissociate at a physiological pH and in presence of a large amount of salts (PBS, TBS) which can indicate a dilution effect.

We appreciate the importance of understanding the mechanism of fibril dissociation in order to control and optimize this process. Even if we would like to discuss further any potential dilution or electrostatic repulsion effects, we believe that a more detailed characterization of the equilibrium between free and fibrillar peptide (to assess k_{on} and k_{off}) is required to draw any conclusions. In a subsequent study (in preparation), we are characterizing the thermodynamics and kinetics of the fibrils by changing peptide concentration, ionic strength, temperature etc.

(ii) The problem of reproducibility of peptide release from the amyloids, the potential presence of polymorphs (see also reviewer 2 remarks), and the administration of the same dose of potentially sticky amyloids have not been resolved adequately and it is thus of concern. This issue is also conceptually important, since the approach undertaken in the published and references GNRH analog studies started mainly with soluble material that aggregated in the body, while the presented

approach incubates amyloids. The reviewer requests therefore again to measure in vitro the reproducibility of the release of hormones after taking defined amounts of de novo prepared hormone amyloids generated by different starting batches. In this context, the Table S1 is cited, which shows actually such data, but it is unclear whether different batches and starting material was used or whether the data come from a single experimental set up.

Reply: As requested, we have now added, in the supplementary Table 1, data showing % release after 4 h incubation in water of 3 independent batches of fibrils (37.8 ± 3.8 shown as mean \pm standard error of mean from three independent batches of fibrillar Oxm and three samples from each batch). These data show *in vitro* reproducibility for peptide release in this condition. We appreciate the concerns raised by the reviewer and the parameters cited will be further optimized (like any other APIs in drug development) to enable dose accuracy.

Reviewer #2 (Remarks to the Author):

In the revised manuscript, the authors added requested information and addressed feasible major issues.

Reply: We thank the reviewer for this comment.

I have one comment to the authors regarding the risks of amyloid-based therapy: I don't agree with the response that "we believe that this risk is minimal as disease-related proteins or peptides forming these aggregates (for instance beta-amyloid peptide, alpha-synuclein and prion) have very different sequences...". Although some publications suggested that sequence similarity is required, many studies showed cross-seeding between remote systems. In any case, I don't think that this is a major issue to be addressed here since this manuscript presents a proof of concept and not an actual drug. The studies of amyloid-based therapy will require many years in order to assess the actual risks.

Reply: We agree with the reviewer of this potential risk which will be further investigated by performing cross-seeding experiment and assessing long-term *in vivo* toxicity.

Reviewer #3 (Remarks to the Author):

The authors have addressed all the issues raised

Reply: We thank the reviewer for this comment.

REVIEWERS' COMMENTS:

Reviewer #1 (Remarks to the Author):

The authors have now resolved the remaining issues including addition of experimental data.